# The Potential Role of News Media to Construct a Machine Learning Based Damage Mapping Framework

**Genki Okada** [1], **Luis Moya** [2,3], **Erick Mas** [2] and **Shunichi Koshimura** [2,*]

1 Graduate School of Engineering, Tohoku University, Aoba 468-1, Aramaki, Aoba-ku, Sendai 980-8572, Japan; genki.okada.p2@dc.tohoku.ac.jp
2 International Research Institute of Disaster Science, Tohoku University, Aoba 468-1, Aramaki, Aoba-ku, Sendai 980-8572, Japan; lmoyah@uni.pe (L.M.); mas@irides.tohoku.ac.jp (E.M.)
3 Japan-Peru Center for Earthquake Engineering Research and Disaster Mitigation, National University of Engineering, Tupac Amaru Avenue 1150, Lima 25, Peru
* Correspondence: koshimura@irides.tohoku.ac.jp

**Abstract:** When flooding occurs, Synthetic Aperture Radar (SAR) imagery is often used to identify flood extent and the affected buildings for two reasons: (i) for early disaster response, such as rescue operations, and (ii) for flood risk analysis. Furthermore, the application of machine learning has been valuable for the identification of damaged buildings. However, the performance of machine learning depends on the number and quality of training data, which is scarce in the aftermath of a large scale disaster. To address this issue, we propose the use of fragmentary but reliable news media photographs at the time of a disaster and use them to detect the whole extent of the flooded buildings. As an experimental test, the flood occurred in the town of Mabi, Japan, in 2018 is used. Five hand-engineered features were extracted from SAR images acquired before and after the disaster. The training data were collected based on news photos. The date release of the photographs were considered to assess the potential role of news information as a source of training data. Then, a discriminant function was calibrated using the training data and the support vector machine method. We found that news information taken within 24 h of a disaster can classify flooded and nonflooded buildings with about 80% accuracy. The results were also compared with a standard unsupervised learning method and confirmed that training data generated from news media photographs improves the accuracy obtained from unsupervised classification methods. We also provide a discussion on the potential role of news media as a source of reliable information to be used as training data and other activities associated to early disaster response.

**Keywords:** disaster; flood; machine learning; training data collection; remote sensing

## 1. Introduction

In recent years, the frequency of typhoons and heavy rains that cause floods has increased in Japan and other countries. When a flood occurs, it is necessary to understand the entire flooded area to search and rescue citizens [1]. It is also essential to identify the extent of the flooded area for vulnerability and risk studies [2–4]. Local governments must understand the details of the damage and prevent the spread of damage and its recurrence [5,6]. For carrying out such rescues and recovery activities efficiently, it is required to assess the damage at each building in the affected area. Remote sensing technology has attracted attention for wide-area damage assessment during disasters. Synthetic Aperture Radar (SAR) image analysis enables us to observe the ground surface without being affected by time and weather conditions [7–9].

A SAR image is produced from an active sensor. Therefore, the electromagnetic waves backscattered from smooth surfaces, such as water bodies, have a clear pattern: their intensity is very low. Thus, in SAR images, flood areas might be associated with areas with low intensity. An additional SAR image recorded before the occurrence of

the flood can be used as a baseline to remove permanent water bodies [10]. However, special attention must be paid to anthropic activities [8]. Identifying changes between pre-event and post-event images is a common approach to identify the effects of many kinds of disasters [11–14]. For the identification of flood-based water bodies, a reduction of the backscattering intensity is expected. It has been pointed out that detecting floods in urban areas is, however, more challenging [15]. In addition to the low backscattering intensity associated with the specular reflection mechanism, a common join effect of specular reflection and double bounce may produce high backscattering intensity [16,17]. Therefore, the change in intensity between the pre- and post-event SAR images can be either positive or negative. In addition to backscattering, other features such as interferometric coherence [8,9,12,18], texture features [19,20], semantic labels [21], multi-sensors data [22], and phase correlation [14] have been used to identify flooded urban areas. The referenced features are generally analyzed using machine learning methods. Other studies use deep neural networks to identify the features inherent in the images [23,24]. To use machine learning methods for early disaster response, a sufficient amount of training data is required. Although it is desirable to prepare training data based on field surveys, it may take several days to several weeks immediately after a disaster. There are some solutions to this problem, the most intuitive of which is the use of training data collected from past events [9,23]. This solution may increase the complexity of the problem. Building samples from the previous disaster might not be in the same spatial domain as in the current disaster, which is reflected by the different distributions of flooded and nonflooded buildings [9]. In [23], data augmentation techniques were performed to overcome this issue. Another solution involves avoiding the use of training data by using different constraints. In [25,26], the use of damage functions developed for disaster risk analysis and numerical models of the disaster were used to calibrate a machine learning classifier.

Recently, information from social networks and news media has been studied for disaster management due to its capacity to disseminate information regarding the threatening disaster and evacuation orders. In [27], the factors influencing people's perception of the cost and benefits of taking recommended protective measures were assessed. The referred study stressed the role of social marketing to reduce the perceived cost and increase the benefits. In [28], Twitter was used as a social sensor to identify geotemporal patterns associated with the disaster, assessing the damage extent. In general, disaster response agencies commonly use social media photographs as essential eyewitness photos in disaster response [29]. News media have more vital and robust resources to disseminate information quickly. Among the resources are:

1.  The staff is prepared to act immediately;
2.  Remote-controlled cameras strategically placed on top of tall buildings and pointing to the society and hazardous areas (i.e., volcanoes, nuclear power plant);
3.  Camera crews that are out to send information, videos, and images;
4.  Helicopters to deploy for coverage of the news;
5.  They operate in satellite and cable network communication systems.

Therefore, news media can broadcast disaster-related information within a few minutes [30]. It is also found that oblique images, such as those taken by news media from helicopters, are useful in disaster response [31].

From the above, the weather news media can be used to geolocate flooded and nonflooded buildings. How precise a machine learning classifier, with calibration from news media-based training data, is able to achieve to detect the flood extent from SAR imagery is a query this study attempts to elucidate.An experimental evaluation is performed, in which the flood induced by the heavy rainfall of 2018 in western Japan is used as a case study. Flooded areas and buildings were identified from news media photographs published within a few hours, 24 h, and 48 h after the floods [32–36]. The support vector machine (SVM) was then used to calibrate a discriminant function, to be used as a classifier of flooded buildings. Considering SVM's usefulness for land surface classification and flood area determination, we used this machine learning method [37–39]. In addition, the flood

map provided by the Geospatial Information Authority of Japan (GSI) is used to contrast our results. The rest of the paper is structured as follows: Section 2 describes the flow of analysis, the study area, and the data used. Section 3 introduces the feature space, the news media-based training data, and the discriminant function. Section 4 reports the performance of the discriminant function in terms of accuracy at different times. Section 5 provides a discussion on the potential collaboration of the news media community and the remote sensing community. Finally, the conclusions are drawn in Section 6.

## 2. Target Area, Data Sets, and Analysis Flow

### 2.1. 2018 Japan Floods

In western Japan, the heavy rains in 2018 induced severe damage over a wide area. A detailed report can be found in The Government's Disaster Prevention White Paper [40], and it is summarized as follows. The weather front, which remained still in northern Japan since 28 June 2018, moved north to Hokkaido in northern Japan on 4 July, and then it moved south to western Japan on 5 July. The typhoon Prapiroon, which approached Japan at about the same time as the weather front, brought a continuous supply of warm and very moist air that resulted in widespread, record-breaking rainfall that led to floods. The heavy rains produced 1800 mm of rainfall to the Shikoku region in southwest Japan and 1200 mm to the Tokai region in eastern Japan, two to four times the average monthly rainfall in July. The heavy rains in western Japan caused extensive damage in a wide area. The areas that were affected the most were the Prefectures of Hiroshima, Ehime, and Okayama. In particular, Okayama Prefecture is characterized by the flood that occurred in Mabi-town, Kurashiki City. A total of 51 casualties were reported. In this study, Mabi-town is selected as the target area. The map of Mabi-town is shown in Figure 1.

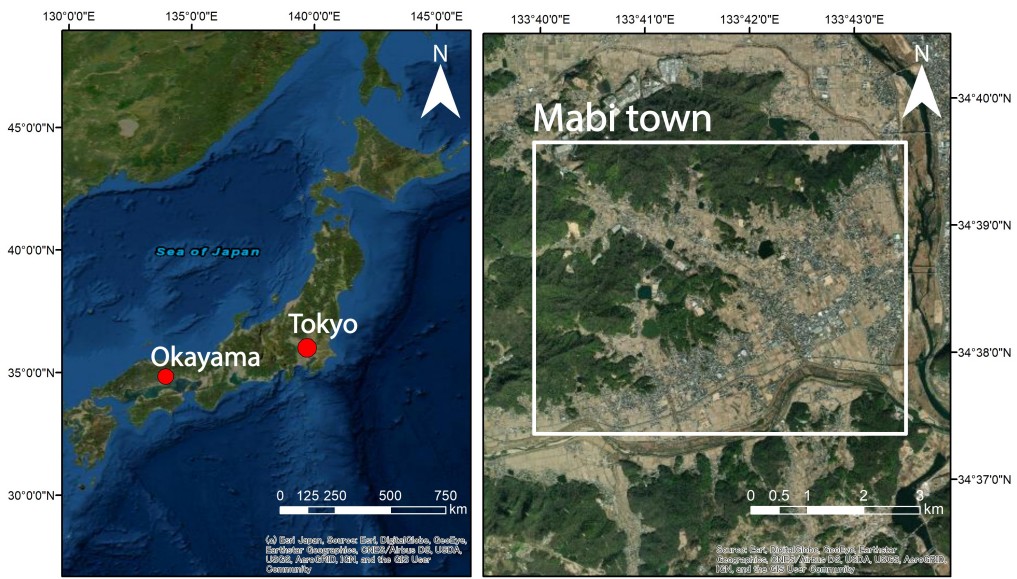

**Figure 1.** Map of Japan and Mabi-town, Okayama Prefecture as a target area.

### 2.2. Land Observation by ALOS-2

The ALOS-2 satellite was developed by the Japan Aerospace Exploration Agency (JAXA) as ALOS's successor. ALOS-2 products have contributed to a wide range of mapping, regional observation, disaster monitoring, and resource exploration applications. ALOS-2 carries the PALSAR-2, an enhanced version of the L-band Synthetic Aperture Radar (PALSAR) onboard ALOS. Unlike the optical sensors, ALOS-2 is not affected by day and night or weather conditions. In the aftermath of the heavy rainfall of 2018, ALOS-2 performed observations in the affected area. Figure 2 shows a SAR image of the affected area recorded on 8 July, Japan Standard Time (JST). A second image recorded on 14 April under similar acquisition conditions is shown as well. The incident angle was about

37 degrees, HH polarization, and a resolution of about 3 m. The images were calibrated, speckle filtered, and terrain corrected with ENVI/SARscape [41].

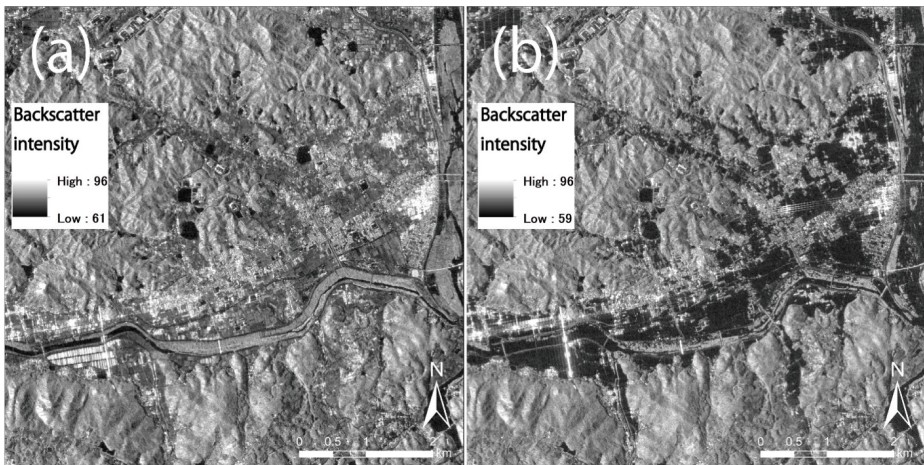

**Figure 2.** SAR image of Mabi-town, (**a**) Data on 14 April 2018 (JST), (**b**) Data on 8 July 2018 (JST).

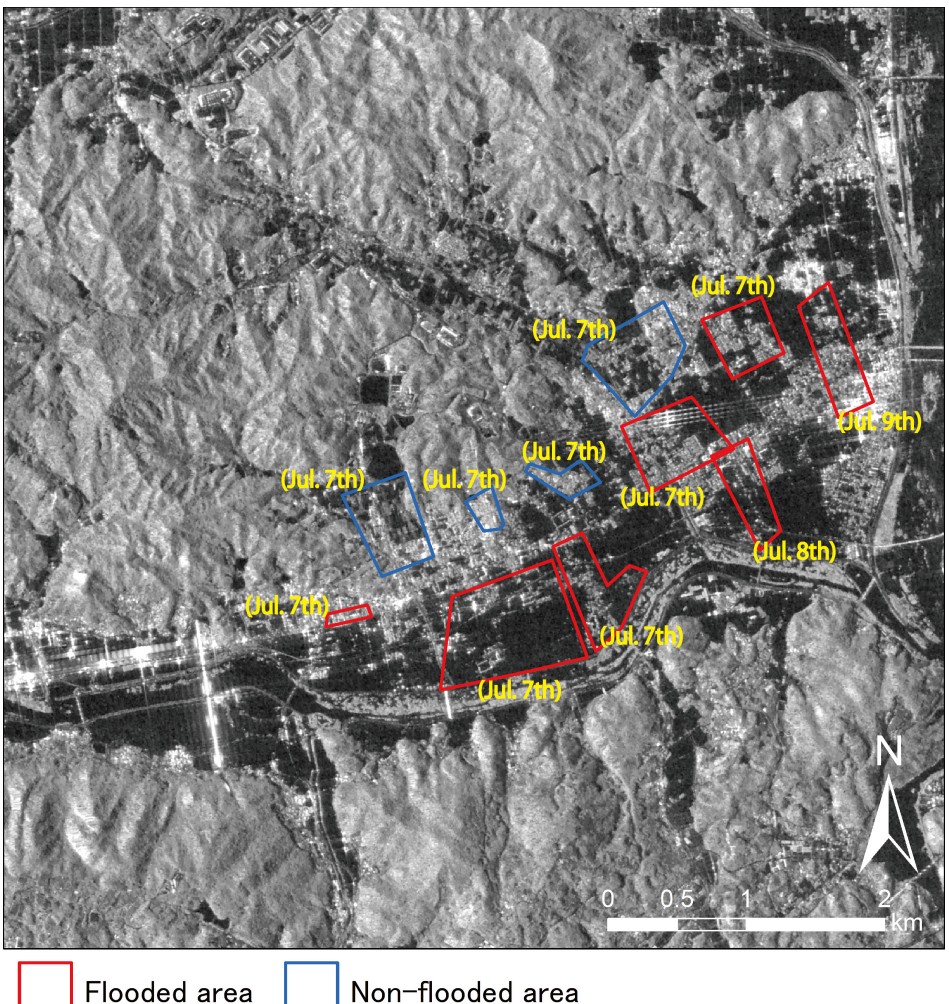

**Figure 3.** Flooded and nonflooded areas interpreted by news media photographs with publication dates.

*2.3. The Flow of Analysis, News-Based Truth Data, and GSI Flood Map*

As mentioned previously, we aim to assess news media-based training data. Flooded areas were identified from The Yomiuri Shimbun (YS) [32], The Asahi Shimbun (AS) [34], Jiji Press Ltd., (JJ) [35], and All-Nippon News Network (ANN) [36]. Table 1 reports the release date of the information. It is observed that YS, MS, and AS provided information by midday of 7 July, JJ reported affected areas on the next day, and ANN published a video of the affected area by 11:48 a.m. on 9 July. In western Japan, a weather front developed from 5 July, but the torrential rains did not start until late in the evening of 6 July. In addition, rivers began to overflow and collapse, and towns began to be inundated from 7 July onward, which is when the news started to report the damage. From the above, we have used news media information after 7 July.

**Table 1.** News media used to create training data and publishing time.

| News Media | Publishing Time |
| --- | --- |
| The Yomiuri Shimbun [32] | 0:33 p.m. on 7 July |
| The Mainichi Newspapers [33] | 0:41 p.m. on 7 July |
| The Asahi Shimbun [34] | 0:49 p.m. on 7 July |
| Jiji Press Ltd. [35] | Afternoon on 8 July |
| All-Nippon News Network [36] | 11:48 a.m. on 9 July |

Based on visual inspection, eleven areas were identified in Mabi-town (See Figure 3), from which seven were flooded and four were nonflooded. Buildings located within the referred areas were collected to calibrate a discriminant function. A total of 1109 buildings were identified in the flooded areas, and 513 buildings were identified in the nonflooded areas. In order to consider the relevance of the time-release of information, three sets are defined. The set $S_1$ contains building samples collected from news published on 7 July. $S_1$ (#$S_1$) cardinality is 1238, which consists of 725 flooded and 513 nonflooded buildings. The set $S_2$ contains building samples collected from news published until 8 July. The #$S_2 = 1579$; that is, 341 additional flooded buildings were identified from JJ news. The set $S_3$ contains building samples collected until 9 July. The #$S_3 = 1622$, which means that only 43 additional buildings were identified from ANN news. Note that $S_1 \subseteq S_2 \subseteq S_3$.

Regarding labeling the training samples, we manually tagged large flooded/nonflooded areas observed in news photographs and translate them into the labels of building footprint data on GIS. It is applied in three steps. First, large areas of flooded or nonflooded, hereafter referred as blocks, are identified from news media photos and videos. Each block contained between 100 and 200 buildings. For the case of flooded block, at least one floor was flooded in all buildings. Second, news media photos that include blocks are pinpointed in satellite images. This step was performed manually for the case study. Landmarks in the photos and addresses reported in the news assisted in their geolocation. Third, building footprints inside the flooded/non-flooded blocks are automatically identified and labeled as flooded/nonflooded. An overview of analysis flow, including generating news media-based training data, is shown in Figure 4.

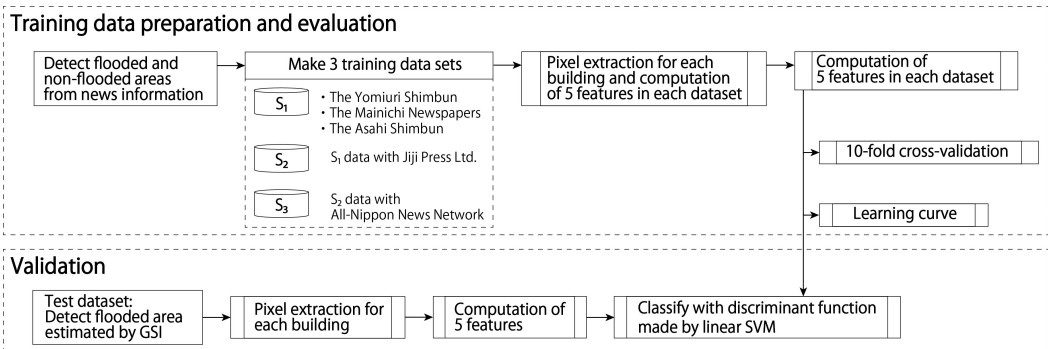

**Figure 4.** Overview of the study.

In addition to the information collected by the news media, the flood map reported by the GSI [42] is used as a material for creating test data to evaluate the discriminant function's performance. The GSI confirmed the flooded area based on aerial images and photographs available from social networking services sources.

## 3. Feature Space and Discriminant Function

A set of hand-engineered features are computed to express the level of similarity between the pair of images. A previous study has shown that interferometric coherence can be used to identify flooded buildings [9,18]. Under normal conditions, urban areas show high coherence, whereas flooded urban areas exhibit low coherence. Regarding the backscattering intensity in urban areas, it mainly shows high values in medium resolution SAR images. On the other hand, high-resolution SAR images show more complex patterns with low values associated with the shadowing effect and large values related to the double bounce backscattering mechanism [43]. Water bodies produced by the floods in urban areas generate modifications in the backscattering intensity. If the flood depth is lower than the building environment, the backscattering intensity increases due to the joined effect of specular reflection and double bounce mechanism [44]. In order to measure the variations in the backscattering intensity, the three-dimensional texture features—contrast, dissimilarity and homogeneity—are employed [20]. The referred hand-engineered features are expressed as follows:

$$\text{Coherence} = \frac{|\sum_{(i,j)} I_{i,j}^{pre} I_{i,j}^{post*}|}{\sqrt{\sum_{(i,j)} I_{i,j}^{pre} I_{i,j}^{pre*} \sum_{(i,j)} I_{i,j}^{post} I_{i,j}^{post*}}} \tag{1}$$

$$\text{Contrast} = \frac{1}{N} \sum_{(i,j)} (|I_{i,j}^{pre}| - |I_{i,j}^{post}|)^2 \tag{2}$$

$$\text{Dissimilatity} = \frac{1}{N} \sum_{(i,j)} ||I_{i,j}^{pre}| - |I_{i,j}^{post}|| \tag{3}$$

$$\text{Homogeneity} = \frac{1}{N} \sum_{(i,j)} \frac{1}{1 + (|I_{i,j}^{pre}| - |I_{i,j}^{post}|)^2} \tag{4}$$

$$\text{Correlation} = \frac{1}{N} \sum_{(i,j)} \frac{(|I_{i,j}^{pre}| - \mu^{pre})(|I_{i,j}^{post}| - \mu^{post})}{\sigma^{pre}\sigma^{post}} \tag{5}$$

where $I_{i,j}^{pre}$ denotes the complex backscattering of the pre-event SAR image, $I_{i,j}^{post}$ denotes the complex backscattering of the post-event SAR image, $*$ denotes the complex conjugate, $|\cdot|$ denotes the complex amplitude, $\mu$ and $\sigma$ denote the mean value and the standard deviation of the backscattering intensity. The features were computed at each sample building collected from the news. When calculating the features, we extracted values from

each building's interior and surrounding pixels and used them, as shown in Figure 5, following the method to detect damaged buildings [20]. Then, each feature was normalized to have zero mean and unit variance. Figure 6 depicts the density distribution of the $S_3$ training data features, which shows the differences between flooded and nonflooded buildings. Figure 6 indicates that the training data can be categorized into flooded and nonflooded buildings.

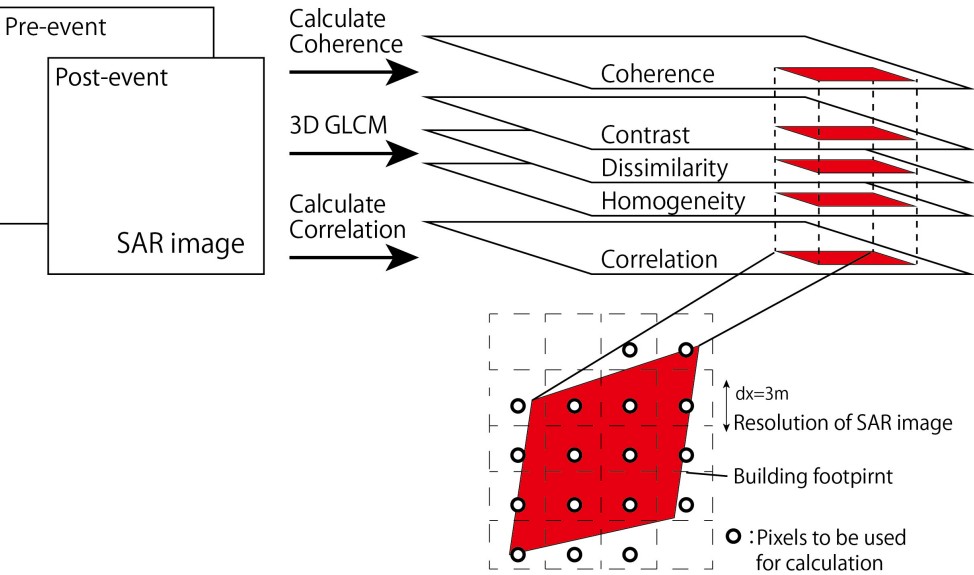

**Figure 5.** The method how to extract value from interior and surrounding pixels of building.

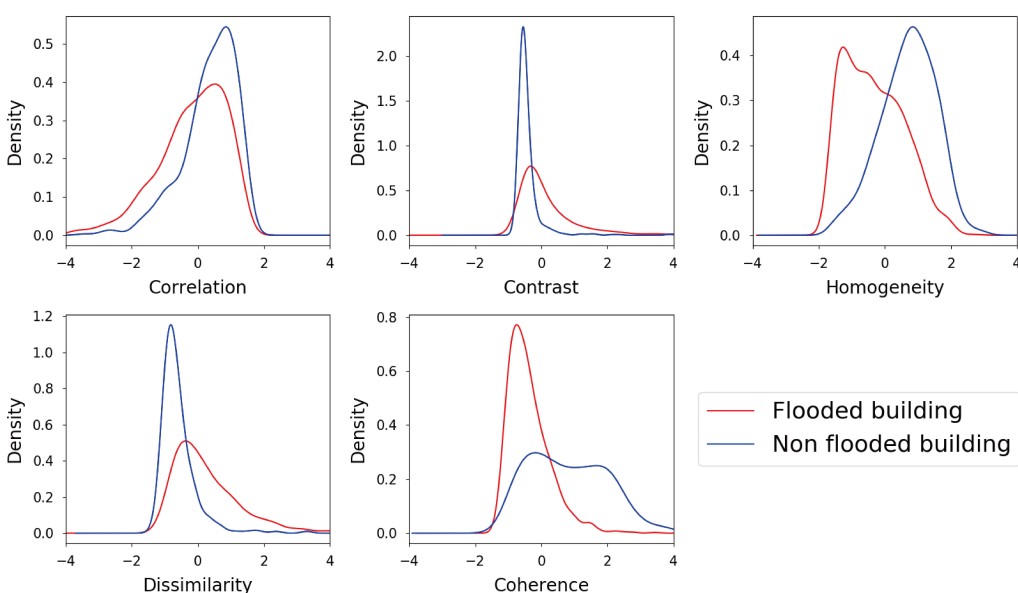

**Figure 6.** Density distribution of the features. Red and blue lines show the flooded and nonflooded buildings, respectively.

As mentioned in the introduction, the SVM method is employed to calibrate the following linear discriminant function [45]:

$$f(x_i) = \text{sign}(w \cdot x_i + \rho) \tag{6}$$

where $x_i$ is a sample vector that contains the features, $w$ is a vector perpendicular to the hyperplane $w \cdot x_i + \rho = 0$; and $\rho$ is a constant offset. If a sample building $x_i$ is flooded,

then $f(x_i) = 1$, otherwise, $f(x_i) = -1$. The parameters $w$ and $\rho$ are calculated from the following optimization problem:

$$\min_{w,\xi} \left\{ \frac{1}{2} w \cdot w + \lambda \sum_{i}^{N} \xi_i \right\} \tag{7}$$

$$\text{subject to } z_i(w \cdot x_i + \rho) - 1 + \xi_i \geq 0$$
$$\xi_i \geq 0$$

where $x_i$ is a training sample, $\xi_i$ is a slack variable, $\lambda$ is a regularization parameter defined in advance, $z_i = 1$ if the training sample $x_i$ belongs to the class flooded; otherwise, $z_i = -1$, $N$ denotes the number of training samples. In this study, a linear SVM was used to classify flooded and nonflooded buildings.

The performance of the discriminant function made from news-media photographs is evaluated using 10-fold cross-validation and the learning curve. Consider a truth data set of size $N$; the 10-fold cross-validation consists of randomly dividing the truth data into ten subsets. Nine subsets are used for training, and the rest one is used for validation. This process is repeated ten times, each with different subset for the validation step. The 10-fold cross validation is performed for different size of $N$. The learning curve shows the relation between N and the resulted scores. The benefits of increasing the amount of truth data are depicted in the learning curve.

In addition to 10-fold cross-validation and the learning curve, we applied unsupervised learning classification to check whether the news media-based training data caused the classifier to experience over-learning or under-learning. As a classic unsupervised learning method, we employed the K-means method. It has been found that it is possible to extract areas where floods have changed the landscape by applying K-means to SAR imagery [46]. First, assuming the $N$ samples are classified into two clusters, two representative points of each class are determined randomly. Second, the distance between each data and the representative point is then calculated, and the data should belong to the cluster to which the representative point with the shorter distance is located. After doing this for all samples, we compute the centroid of the data classified into each cluster. With this centroid as the new representative point, we reclassify each data set. Finally, this process is repeated until the centroid no longer moves. 10-fold cross-validation, Learning curve, K-means method and Linear SVM were performed with Python language programming using the scikit-learn library [47].

## 4. Results

Figure 7 shows the learning curve for the three sets $S_1$, $S_2$, and $S_3$. The red and green solid line denotes the learning curve computed with the training and testing data, respectively. The shaded areas denote the region within the mean plus/minus the standard deviation of the score. As a general trend, the standard deviation decreases as the number of samples increases, and the mean value shows convergence. For the case of $S_1$, the mean score does not change significantly. Furthermore, there is a difference of about 2% between the score of the training and testing data sets. The results for the set $S_2$ are similar to that from $S_3$. That is, the training data score is high at the beginning and decreases as the number of samples increases. Furthermore, the test data score is low initially and increases as the number of samples increases. It is also observed that the convergence value for the training and testing data sets are similar, and the standard deviation is lower than that computed from the set $S_1$. These suggest that the discriminant function did not incur in over-learning.

Table 2 shows a prediction on testing data from the 10-fold computed using $S_3$, and Table 3 reports the recall, precision, and F1 scores. The recall is the percentage of flooded (nonflooded) buildings predicted to be flooded (nonflooded). Likewise, the precision represents the percentage of buildings predicted to be flooded (nonflooded) that were flooded (nonflooded). The F1 score is calculated from the recall and precision:

F1 $= 2/(\text{recall}^{-1} + \text{precision}^{-1})$. The average scores range from 0.81 to 0.87. These scores indicate a good agreement between the predictions and the truth data. In the same way, as in Table 2, the results of K-means method for each of the 10 groups of data set are shown in Table 4. The recall, precision, and F1 scores calculated from Table 4 are shown in Table 5. Comparing Tables 3 and 5, we can check that the F1 score's average is 0.15 higher for the SVM classification results than for the K-means classification results. In particular, the recall for the classification of flooded buildings and the precision for the classification of nonflooded buildings can be improved by using training data and supervised learning. The results show that applying news media-based training data and SVM effectively identifies flooded and nonflooded buildings with limited information.

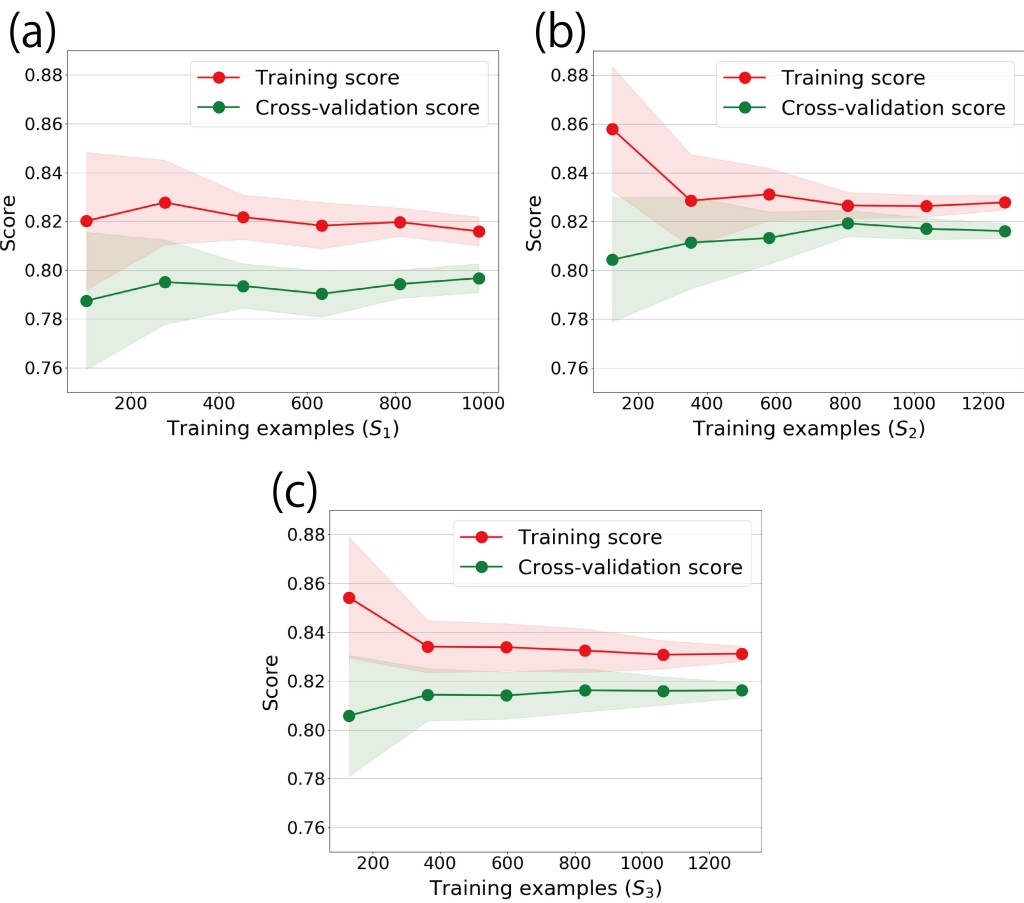

**Figure 7.** The learning curve, (**a**) $S_1$ (Training data on 7 July), (**b**) $S_2$ (Training data until 8 July), (**c**) $S_3$ (Training data until 9 July).

**Table 2.** Result of prediction on $S_3$ data using 10-fold cross-validation by support vector machine (SVM).

| | | Prediction Result | | |
|---|---|---|---|---|
| | | **Flooded Building** | **Non-Flooded Building** | **Total** |
| Truth | Flooded building | 105 | 5 | 110 |
| | Non-flooded building | 17 | 35 | 52 |
| | Total | 122 | 40 | 162 |

**Table 3.** Scores of prediction on SVM, calculated by the Table 2.

|  | Recall | Precision | F1 |
|---|---|---|---|
| Flooded building | 0.86 | 0.95 | 0.91 |
| Non-flooded building | 0.88 | 0.67 | 0.76 |
| Average | 0.87 | 0.81 | 0.83 |

These results suggest that about 80% can be achieved with information gathered from $S_1$, available in near-real-time. Besides, an improvement of about 2% was able to achieve with additional information collected in the next 24 h. The data collected on 9 July did not influence the performance of the classifier. Note that the data collected on 8 and 9 July is small compared with that collected on 7 July, which might be the reason for a small improvement in the scores. In addition to the evaluation, Figure 8 shows the results of the buildings' classification by the discriminant function inside the GSI's estimated flooded area. The red and blue marks denote the flooded and nonflooded buildings, the green marks are the buildings used as training data, and the yellow line denotes the flooded area delineated by GSI. Table 6 reports the predictions performed by the discriminant function calibrated with $S_1$, $S_2$, and $S_3$. It is only possible to compute the recall score, which indicates the percentage of flooded buildings correctly judged to be flooded. The results are consistent with the learning curve; that is, there is an improvement when data collected on 8 July is used.

**Table 4.** Result of prediction on $S_3$ data using 10-fold cross-validation by K-means method.

|  |  | Prediction Result | | |
|---|---|---|---|---|
|  |  | Flooded Building | Non-Flooded Building | Total |
| Truth | Flooded building | 64 | 46 | 110 |
|  | Non-flooded building | 3 | 49 | 52 |
|  | Total | 67 | 95 | 162 |

**Table 5.** Scores of prediction on K-means method, calculated by the Table 4.

|  | Recall | Precision | F1 |
|---|---|---|---|
| Flooded building | 0.96 | 0.58 | 0.72 |
| Non-flooded building | 0.48 | 0.94 | 0.64 |
| Average | 0.72 | 0.81 | 0.68 |

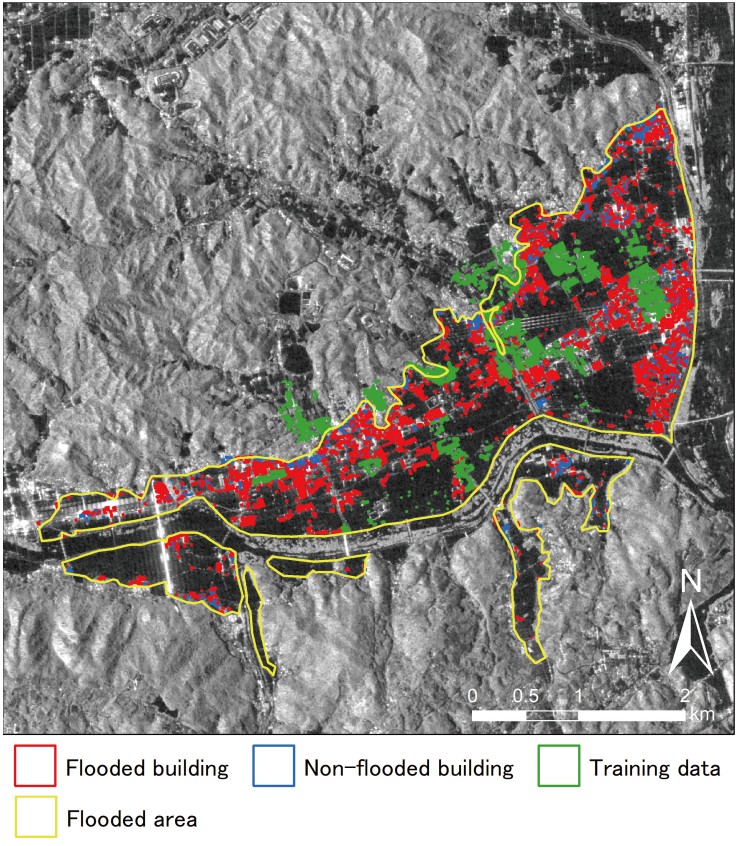

**Figure 8.** SVM estimating result using $S_3$ training data.

**Table 6.** Classification result for buildings inside of Geospatial Information Authority of Japan's (GSI's) estimated flooded area. The total number of buildings: 4391.

| Source of Training Data | Flooded Building | Nonflooded Building | Recall (%) |
|:---:|:---:|:---:|:---:|
| $S_1$ | 3119 | 1272 | 71.0 |
| $S_2$ | 3603 | 788 | 82.1 |
| $S_3$ | 3642 | 749 | 82.9 |

## 5. Discussion

The results show that the news media information is useful for gathering reliable training data, which enables one to detect the whole extent of the flooded area with the aid of machine learning techniques. Although the news media-based training data is fragmented, it is effective enough for supervised learning classification purpose. Regarding the number of training samples, we estimate that it is sufficient to identify 500 to 1000 buildings for each class, flooded and nonflooded buildings.

In addition to the quantitative evaluation, we found necessary additional points regarding the characteristics of news-data that can contribute to identify the extent of a flooded area. We consider four essential aspects that news photographs should consider in order to be used together with remote sensing data for early disaster response. First, information regarding flood depth can be estimated from the building's sidewalls. Side looking imagery are useful in this regard. In a previous study, aerial photographs taken from a helicopter were used to assess damage to buildings that had collapsed or cracked due to earthquakes. It was noted that the photographic angle of 30–45 degrees was the right balance in assessing building damage [48]. For the explicit case study reported in our study, the Yomiuri Shimbun's photographs clearly showed the inundation depth were about the first floor height in some buildings. Such information contributes to a understand the magnitude of the damage. Second, recording time of the photography must

be available. For example, The Asahi Shimbun published the photo's recording time [34]. It has been pointed out that the difference between the acquisition time of the SAR images and that from the photographs used as training data reduces the analysis's accuracy [10]. Furthermore, the chronological changes, such as contraction or expansion of the affected area, can be estimated if news media photographs with recording time is available. This is useful to confirm the gap between the SAR-based flood map and that from visual inspection of photographs [9]. Third, photographs should be geocoded and contain landmarks. An important pitfall is that, in most cases, news does not provide geocoded photographs. In the case study the photographs were geocoded manually. However, sharing geocoded photographs would represent a step further into a potential collaboration between news media community and the technicians that use remote sensing data for damage mapping. Furthermore, in order to associate the buildings observed in the photographs to a geocoded building footprint database, landmarks such as schools, hospitals, high-rise buildings, rivers, stations, and shopping malls should be included in the news photographs. Manual tagging of flooded/nonflooded will be more efficient if the news photos capture these local landmarks. Fourth, nonflooded urban areas near the affected area should also be reported. This is because the calibration of a machine learning classifier requires training data of both flooded and nonflooded samples.

Damage mapping is extremely useful for early disaster response, which consists of all the activities required to save lives, alleviate suffering, and facilitate rescue operations. It is expected that the referred activities must be performed within the first 72 h [49]. We believe training data can be obtained from news media information within a few hours, because training data collection using news media photos is done manually but in block scale. Another important factor that affects the release date of damage maps is the acquisition date of the remote sensing data. However, both tasks are not in conflict because they can be performed in parallel. Therefore, a collaboration between news-media and remote sensing communities will provide timely and accurate damage maps.

## 6. Conclusions

In this study, we assessed news media as a source of information to prepare flood damage maps using machine learning. We aimed to study the level of accuracy at different timeline stages of a machine learning classifier calibrated from training data collected from news media photographs. The flood occurred in Mabi-town, Japan, and was selected as a study case, from which news photographs/videos were collected. The news media photographs were used to geolocate flooded and nonflooded buildings. It is worth noting that about 76% (1238) of the building damage inventory we collected from information released within a few hours after the occurrence of the flood, about 19% buildings from the next 24 h, and about 3% on the second day after the flood. The buildings captured in news media photographs were then used as training data to identify change patterns from a pair of SAR images in the building environment. As a machine learning for flooded building detection, the support vector machine was used to calibrate a classifier.

The evaluation of the classifier calibrated from news media-based training data was performed using 10-fold cross-validation, the learning curve, and a comparison with an unsupervised classifier. The results show that with the information used in the first hours ($S_1$) provided about 80% accuracy, and the accuracy increased up to about 82% with data collected in the following 24 h ($S_2$). The reason for the improved accuracy could be due to the fact that $S_2$ and $S_3$ contain $S_1$ and thus more data was learned. Besides, the results were consistent with the flood map published by the GSI. Overall, the results indicate that the news media photographs are beneficial for generating flood maps in near-real-time. Furthermore, we provided a set of recommendations to the news media community to collaborate with the remote sensing community for early disaster response activities.

**Author Contributions:** Conceptualization, G.O., L.M., E.M. and S.K.; methodology, G.O., L.M., E.M. and S.K.; software, G.O., L.M.,E.M. and S.K.; validation, G.O., L.M., E.M. and S.K.; formal analysis, G.O., L.M., E.M. and S.K.; investigation, G.O., L.M., E.M. and S.K.; resources, G.O., L.M.,

E.M. and S.K.; data curation, G.O., L.M., E.M. and S.K.; writing—original draft preparation, G.O., L.M., E.M. and S.K.; visualization, G.O., L.M., E.M. and S.K.; supervision, L.M., E.M. and S.K.; project administration, G.O., L.M., E.M. and S.K.; funding acquisition, G.O., L.M., E.M. and S.K. All authors have read and agreed to the published version of the manuscript.

**Funding:** This work was funded by Japan Society for the Promotion of Science (JSPS) Kakenhi (17H06108); the Concytec-World Bank project No. 8682-PE through its executing unit Fondecyt (contract number 038-2019).

**Acknowledgments:** This research was partly supported by Japan Aerospace Exploration Agency (JAXA), and Tough Cyber-physical AI Research Center, Tohoku University, the Core Research Cluster of Disaster Science at Tohoku University (a Designated National University). The satellite images were provided by Japan Aerospace Exploration Agency (JAXA) and preprocessed with ArcGIS 10.6 and ENVI/SARscape 5.5, and the other processing and analysis steps were implemented in Python using GDAL and NumPy libraries.

**Conflicts of Interest:** The authors declare no conflict of interest.

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
