# Peer review of "The Potential Role of News Media to Construct a Machine Learning Based Damage Mapping Framework"

_remotesensing, doi:10.3390/rs13071401_

Round 1

Reviewer 1 Report

This is a clear, concise, and well-written manuscript. The author proposed a news media source-based training dataset construction method to improve remote damage detection. The idea is good and the whole framework is intact. But, there are some issues associated with Publication.

  1. The author proposes the news media-enabled training dataset construction, so I think the author should add a section to describe how to quantitatively transfer the news media to the dataset labels.
  2. Surely, the performance of supervised learning is better than unsupervised learning as the conclusion in the manuscripts, But for early disaster response, do we have enough time to label the new dataset? How to change the news media to a training dataset in real-time?

Reviewer 2 Report

1. It seems that authors use mass media data in order to analyse data visually and to create map of flooded/non-flooded buildings in order to use it in further training procedure. In such a case, for what purpose they require information regading flood depth (see lines 263-264). They did not use such an information in their study.
2. They state that mass media information used in the first hours provided 78.8% accuracy, and the accuracy increased up to 82% with the data collected in the following 24 hours (lines 304-306). Why? This statement should be commented.
3. What will be an accuracy of flooded buildings discrimination using just hand-engineered features from (1)-(5), without any external information like as mass media data?

Reviewer 3 Report

The proposed paper is aligned with the aims and objectives of the Remote Sensing journal. This paper investigates the potential integration of images derived by news media on a SAR processing workflow to detect flooded building and map flooded areas. The structure of this paper is consistent with the journal guidelines. The bibliography section is also adequate since contemporary papers from world-renowned journals have been cited throughout the text. Nevertheless, I have some reservations regarding the justification of the proposed methodology as a means for early disaster response. I have the following questions/ remarks:

  1. According to the authors, the weather front affected western Japan on July 5th. The news begun broadcasting information on July 7th and went on for two more days. It appears as if there is an at least two-day gap between the manifestation of the heavy rainfall and the actual provision of footage from the news media. Is this the case?
  2. Please, state the software(s) that you used to assess the proposed methodology? In case you developed your own code, state the details.
  3. The formation of the dataset was based on visual inspection of 1622 buildings. How long did the acquisition and the identification of the information from the news media take?
  4. As stated in the discussion section, the number of training samples required for the application of the proposed methodology are between 1000 – 2000 buildings (for both classes). As the authors, state, the image geolocation requirement impedes the process. These facts combined with the satellite revisiting time will most likely hinder the accuracy of products and the component of timely response. Therefore, in what specific aspects the proposed methodology justifies a means for ‘early disaster response’?

Round 2

Reviewer 1 Report

The author has revised the manuscript carefully. The methods seem more integrity now. I recommend it for publication.

Reviewer 3 Report

I have no further comments. The answers provided from the authors were concise.